**DOI: 10.1038/ncomms11736**　　**OPEN**

# Quantum teleportation from light beams to vibrational states of a macroscopic diamond

P.-Y. Hou[1], Y.-Y. Huang[1], X.-X. Yuan[1], X.-Y. Chang[1], C. Zu[1], L. He[1] & L.-M. Duan[1,2]

With the recent development of optomechanics, the vibration in solids, involving collective motion of trillions of atoms, gradually enters into the realm of quantum control. Here, building on the recent remarkable progress in optical control of motional states of diamonds, we report an experimental demonstration of quantum teleportation from light beams to vibrational states of a macroscopic diamond under ambient conditions. Through quantum process tomography, we demonstrate average teleportation fidelity $(90.6 \pm 1.0)\%$, clearly exceeding the classical limit of 2/3. The experiment pushes the target of quantum teleportation to the biggest object so far, with interesting implications for optomechanical quantum control and quantum information science.

[1] Center for Quantum Information, Institute for Interdisciplinary Information Sciences, Tsinghua University, Beijing 100084, China. [2] Department of Physics, University of Michigan, Ann Arbor, Michigan 48109, USA. Correspondence and requests for materials should be addressed to L.-M.D. (email: lmduan@umich.edu).

Quantum teleportation has found important applications for realization of various quantum technologies[1–4]. Teleportation of quantum states has been demonstrated between light beams[5–8], trapped atoms[9–12], superconducting qubits[13], defect spins in solids[14] and from light beams to atoms[15,16] or solid-state spin qubits[17,18]. It is of both fundamental interest and practical importance to push quantum teleportation towards more macroscopic objects. Observing quantum phenomenon in macroscopic objects is a big challenge, as their strong coupling to the environment causes fast decoherence that quickly pushes them to the classical world. For example, quantum coherence is hard to survive in mechanical vibration of macroscopic solids, which involves collective motion of a large number of strongly interacting atoms. Despite this challenge, achieving quantum control for the optomechanical systems becomes a recent focus of interest with remarkable progress[19–30]. This is driven in part by the fundamental interest and in part by the potential applications of these systems for quantum signal transduction[25–27], sensing[19] and quantum information processing[19–21]. There are typically two routes to achieve quantum control for the optomechanical systems: one needs to either identify some isolated degrees of freedom in mechanical vibrations and cool them to very low temperature

to minimize their environmental coupling[19,28–30], or use the ultrafast laser technology to fast process and detect quantum coherence in such systems[20–24]. A remarkable example for the latter approach is provided by the optomechanical control in macroscopic diamond samples[20,21], where the motions of two separated diamonds have been cast into a quantum-entangled state[20].

In this paper, we report an experimental demonstration of quantum teleportation from light beams to the vibrational states of a macroscopic diamond sample of $3 \times 3 \times 0.3 \, \text{mm}^3$ in size under ambient conditions. The vibration states are carried by two optical phonon modes, representing collective oscillation of over $10^{16}$ carbon atoms. To facilitate convenient qubit operations, we use the dual-rail representation of qubits instead of the single-rail encoding used in the previous experiments[20–23] and generate entanglement between the paths of a photon and different oscillation patterns of the diamond represented by two phononic modes. Using quantum state tomography, we demonstrate entanglement fidelity of $(81.0 \pm 1.8)\%$ with the raw data and of $(89.7 \pm 1.2)\%$ after the background noise subtraction. Using this entanglement, we prepare arbitrary polarization states for the photon and teleport these polarization states to the phonon modes, with the Bell measurements on the polarization and

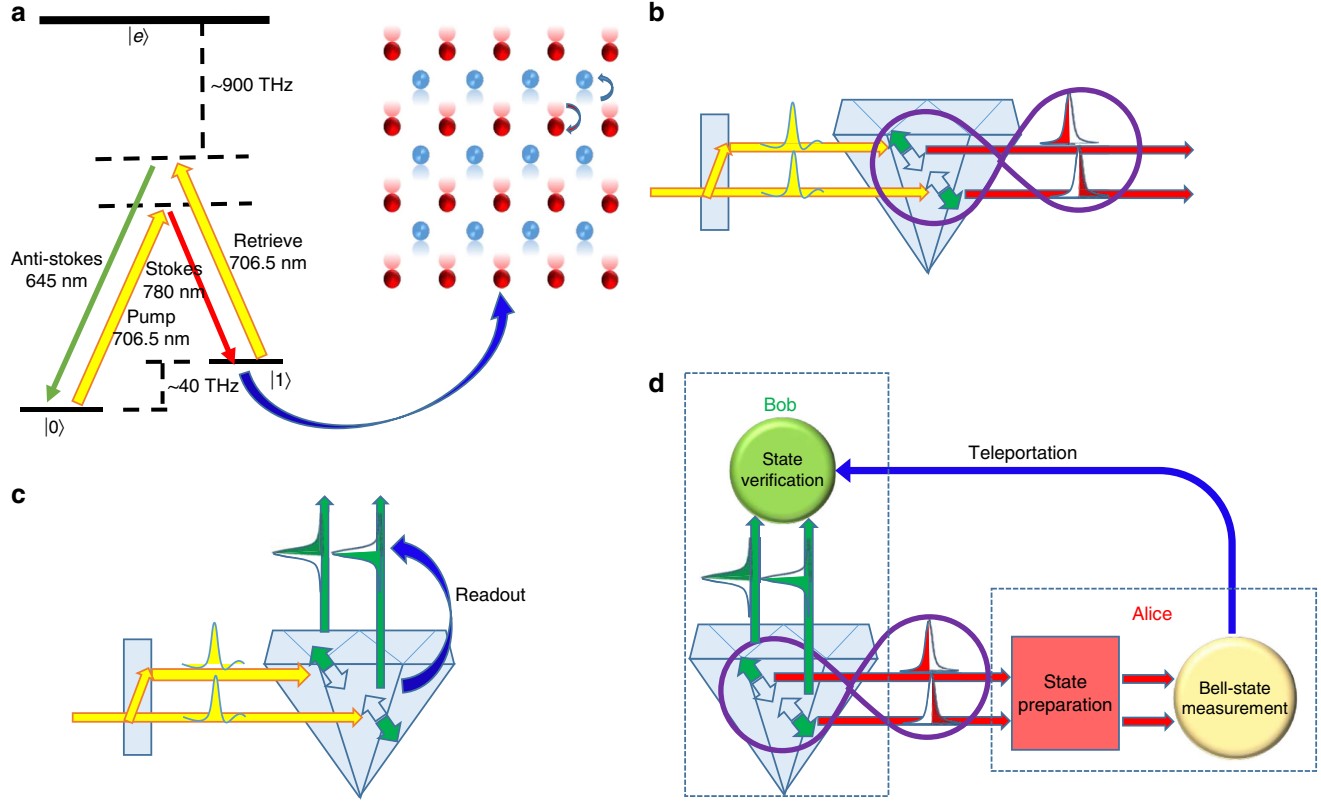

**Figure 1 | Scheme for quantum teleportation with a diamond.** (**a**) Illustration of the relevant level structure in the diamond. A write beam pumps the diamond in the ground state $|0\rangle$ and generates a Stokes photon in the forward direction and an excitation in the optical phonon mode of the diamond (denoted by the state $|1\rangle$). The optical phonon mode corresponds to the relative oscillation of the atoms in each unit cell of the diamond lattice, as illustrated by the figure on the right side. A read beam after a controllable delay converts the phonon excitation to an anti-Stokes photon that can be used for state readout. The corresponding wavelengths and frequencies are shown in the figure. The state $|e\rangle$ denotes the electron conduction band that is far detuned from the optical excitation. (**b**) A scheme for generation of the entanglement between a phonon in the diamond and a propagating photon. The phonon state is represented by a superposition of different oscillation modes of the diamond, while the photon state is represented by its spatial modes. (**c**) Readout of the phonon state with the read beams by coherently converting the phonon modes into the corresponding anti-Stoke photon modes. (**d**) A teleportation scheme using the photon–phonon entanglement. An input state is prepared by the message sender, Alice, on the photon's polarization degree of freedom. The photon thus carries two qubits, one by its polarization and one by its spatial modes. Alice performs Bell measurements on these two qubits. Conditional on certain measurement outcomes, the phonon state is projected to the same state input on Alice's side, which is read out and verified by Bob, the message receiver.

the path qubits carried by the same photon. The teleportation is verified by quantum process tomography (QPT), and we achieve a high average teleportation fidelity, $\sim(90.6 \pm 1.0)\%$ (or $(82.9 \pm 0.8)\%$) after (or before) subtraction of the background noise. To verify the phonon's state before its fast decay, our implementation of teleportation adopted the technique of reversed time ordering introduced in ref. 20 where the phonon's state is read out before the teleportation is completed. Similar to the pioneering teleportation experiment of photons[5], our implementation of teleportation is conditional, as the Bell measurements are not deterministic and require postselecting of successful measurement outcomes.

## Results

**Photon-to-phonon teleportation scheme.** We illustrate our entanglement generation and quantum teleportation scheme in Fig. 1, using a type IIa single-crystal synthetic diamond sample cut along the 100 face from the Element Six company. Due to the strong interaction of atoms in the diamond, the optical phonon mode, which represents relative oscillation of the two sublattices in the stiff diamond lattice (Fig. 1a), has a very high excitation frequency $\sim40$ THz near the momentum zero point in the Brillouin zone. The corresponding energy scale for this excitation is significantly higher than the room temperature thermal energy ($\sim6$ THz), and thus the optical phonon mode naturally stays at the vacuum state under ambient conditions, which simplifies its quantum control[20,21]. The coherence life time of the optical phonon mode is $\sim7$ ps at room temperature, which is short, but accessible with the ultrafast laser technology for which the operational speed can be up to $\sim10$ THz (refs 20,21).

We excite the optical phonon modes through ultrafast laser pulses of duration $\sim150$ fs from the Ti–sapphire laser, with the carrier wavelength at 706.5 nm. The diamond has a large bandgap

of 5.5 ev, so the laser pulses are far detuned from the conduction band with a large gap $\sim900$ THz. Each laser pulse generates, with a small probability $p_s$, an excitation in the optical phonon mode and a Stokes photon of wavelength 780 nm in the forward direction (Fig. 1a). The relevant output state has the form

$$|\Psi\rangle = \left[1 + \sqrt{p_s}b_n^\dagger a_t^\dagger + o(p_s)\right]|vac\rangle, \quad (1)$$

where $b_n^\dagger$ and $a_t^\dagger$ represent, respectively, the creation operators for an optical phonon and a Stokes photon, and $|vac\rangle$ denotes the common vacuum state for both the photon and the phonon modes.

To generate entanglement, we split the laser pulse into two coherent paths as shown in Fig. 1b, and the pulse in each path generates the corresponding phonon–photon correlated state described by equation 1. When there is an output photon, in one of the two paths, it is in the following maximally entangled state with the phonon excitation

$$|\Psi_{nt}\rangle = \left(|U\rangle_n|U\rangle_t + |L\rangle_n|L\rangle_t\right)/\sqrt{2}. \quad (2)$$

Here $|U\rangle$ or $|L\rangle$ represents an excitation in the upper or lower path, and its subscript denotes the nature of the excitation, 'n' for a phonon and 't' for a photon. We drop the vacuum term in equation (1), as it is eliminated if we detect a photon emerging from one of the two paths. After entanglement generation, the photon state can be directly measured through single-photon detectors. To read out the phonon state, we apply another ultrafast laser pulse after a controllable delay within the coherence time of the optical phonon mode, and convert the phononic state to the same photonic state in the forward anti-Stokes mode at the wavelength of 645 nm (Fig. 1c). The state of the anti-Stokes photon is then measured through single-photon detectors together with linear optics devices. Note that the retrieval laser pulse could have a carrier frequency $\omega_r$ different from that of the

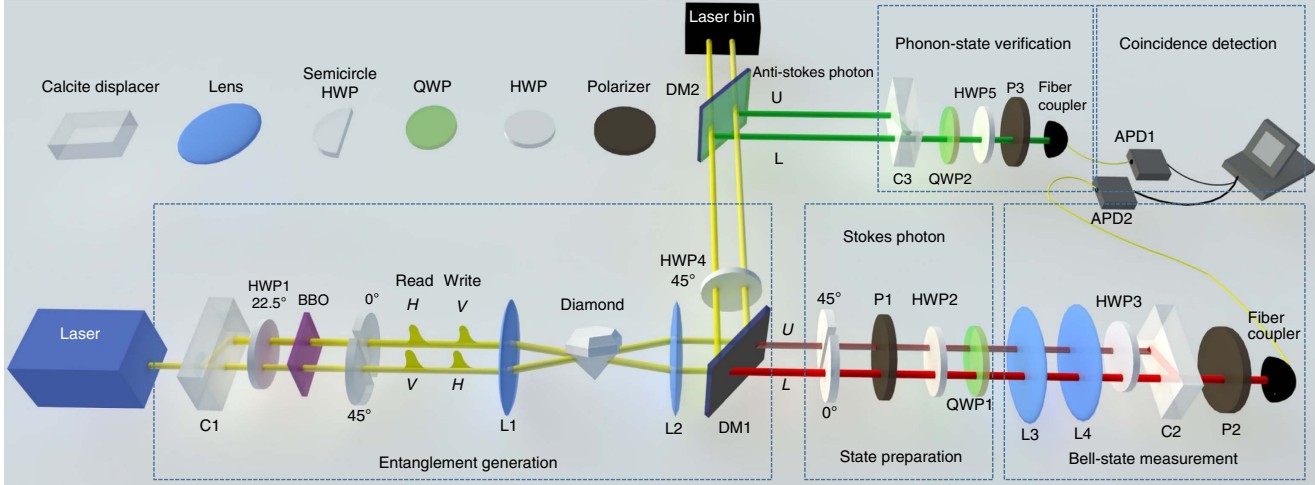

**Figure 2 | Experimental set-up for the entanglement verification and quantum teleportation.** Femtosecond laser pulses from a Ti–sapphire laser (coherent), with a repetition frequency of 76 MHz, a carrier wavelength of 706.5 nm and a polarization along the $|H\rangle + |V\rangle$ direction, are split by a birefringent calcite into two coherent paths with equal amplitudes. After rotation of the pulse polarization to equal superposition of $|H\rangle$ and $|V\rangle$ with a half-wave plate (HWP1) set at 22.5°, we introduce a time delay of 388 fs to the two polarization components H and V, with a birefringent beta barium borate (BBO) crystal. We use the lead pulse of H polarization as the write beam and the lagged pulse of V polarization as the read beam. After semicircle HWPs set at 0° and 45°, respectively, at the upper and lower paths, the polarization states of the pump beams are shown in the figure before the diamond sample. The write beam is focused by the lens L1 on the diamond sample and generates a Stokes photon in one of the paths, and an excitation in the corresponding optical phonon modes of the diamond. The Stokes photon, at the wavelength of 780 nm, is transmitted by the dichromatic mirror DM1 after the collection lens L2, with its two paths recombined by the calcite C2. The lens L3 and L4 are used to adjust the distance between the two optical paths, so that they can be combined at the calcite C2. The single-photon detector APD2, together with rotation of the polarizer P2, detects the two path (or polarization) components of the Stokes photon in different bases. To read out the state of the phonon modes, the read pulse converts the phonon to the anti-Stokes photon in the corresponding paths. The anti-Stokes photon, at a shorter wavelength of 645 nm, is reflected by both of the dichromatic mirrors DM1 and DM2, with its two paths recombined through the calcite C3. The photon coincidence counts are registered through a FPGA (Field-Programmable Gate Array) board with a 5 nm coincidence window.

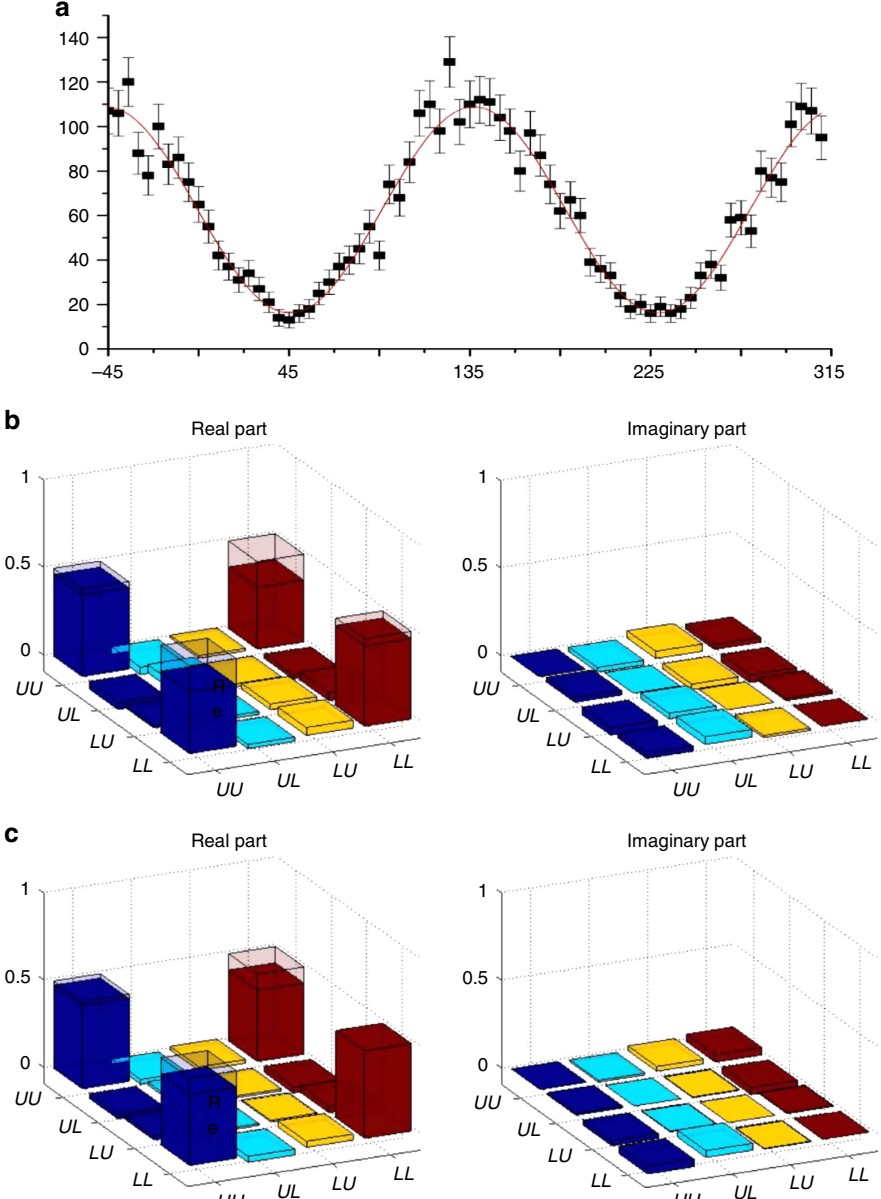

**Figure 3 | Verification of the photon–phonon entanglement.** (**a**) Coincidence counts of Stokes and anti-Stokes photons, as a function of the rotation angle (in degree) of the polarizer (P2 in Fig. 2) for the Stokes photon when the measurement basis of the anti-Stokes photon is fixed at $|U\rangle - |L\rangle$. The error bars denote the s.d. (**b**) Real and imaginary parts of the density matrix elements for the phonon–photon-entangled state reconstructed through the quantum state tomography. The hollow caps correspond to the values of matrix elements for a perfect maximally entangled state. (**c**) Same as **b**, but we subtract the background noise due to the accidental coincidences of the photon detectors. The coincidence count rate for Stokes and anti-Stokes photons is 8 per s for measurements in the $UU$ and $LL$ bases.

pump laser. For instance, with $\omega_r$ near the telecom band, our teleportation protocol would naturally realize a quantum-frequency transducer that transfers the photon's frequency to a desired band, without changing its quantum state. A quantum-frequency transducer is widely recognized as an important component for realization of long-distance quantum networks[25–27].

To realize teleportation, we need to prepare another qubit, whose state will be teleported to the phonon modes in the diamond. Similar to the teleportation experiments in refs 6,16, we use the polarization state of the photon to represent the input qubit, which can be independently prepared into an arbitrary state $c_0|H\rangle_t + c_1|V\rangle_t$, where $|H\rangle_t$ and $|V\rangle_t$ denote the horizontal and the vertical polarization states and $c_0$, $c_1$ are arbitrary

coefficients. The Bell measurements on the polarization and the path qubits carried by the same photon can be implemented through linear optic devices together with single-photon detection (Fig. 1d), and the teleported state to the phononic modes is retrieved and detected through its conversion to the anti-Stokes photon. Same as ref. 20, the short life time of the diamond's vibration modes requires us to retrieve and detect the phonon's state before applying detection on the Stokes photon, thus the phonon's state is measured before the teleportation protocol is completed. The reversed time ordering in this demonstration of quantum teleportation makes it unsuitable for application in quantum repeaters that requires a much longer memory time; however, it does not affect application of our teleportation experiment for realization of a

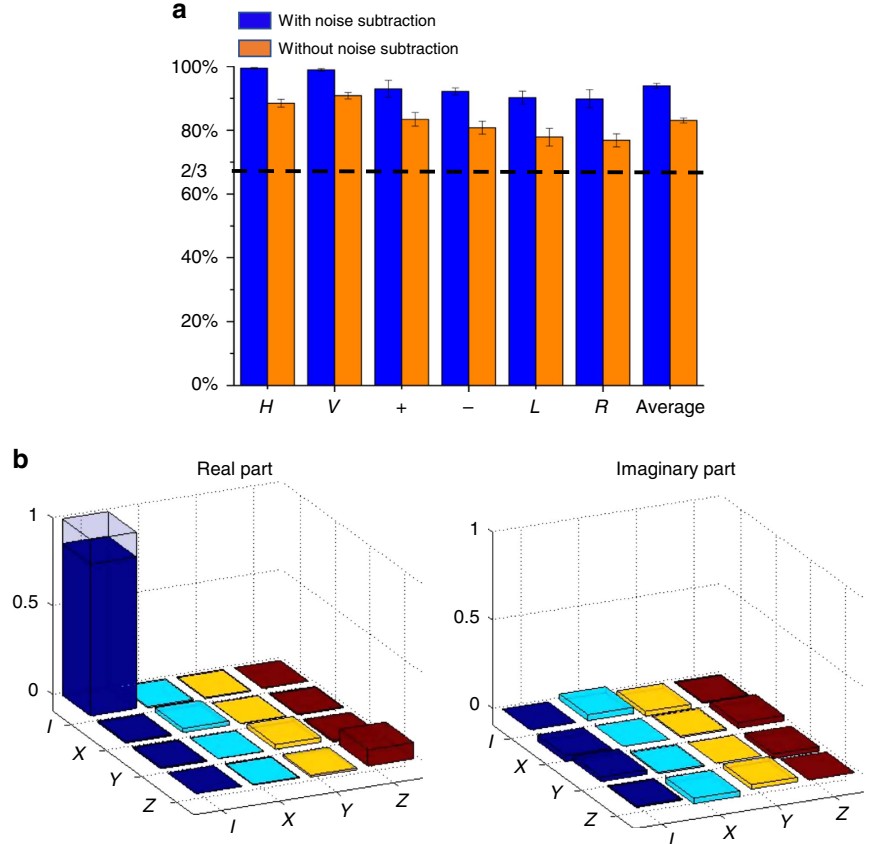

**Figure 4 | Experimental results for photon-to-phonon quantum teleportation.** (**a**) The teleportation fidelities for the six complementary bases states. The last two columns show the teleportation fidelity averaged over these six input states. The results are shown for both cases with or without subtraction of the background noise. The error bar denotes one s.d. The dashed line at fidelity 2/3 corresponds to the classical-quantum boundary for teleportation. (**b**) Real and imaginary parts of the process matrix elements for quantum teleportation reconstructed through the quantum process tomography (Methods). The hollow caps correspond to the values of process matrix elements for a perfect teleportation operation.

quantum-frequency transducer or a new source of entangled photons as discussed above.

**Experimental realization of teleportation.** Our experimental set-up is shown in Fig. 2. First, we verify entanglement generated between the Stokes photon and the optical phonon modes in the diamond. For this step, we remove the optical elements in the state preparation box shown in Fig. 2 and set the angle of half-wave plate (HWP3) to 0°. Different from the scheme illustrated in Fig. 1, we insert semicircle HWPs set at 0° and 45°, respectively, at the upper and the lower paths of the pump beam, so that both the Stokes photon and the anti-Stokes photon after the retrieval pulse have orthogonal polarizations along the two output paths, which can be combined together through the calcites C2 and C3. This facilitates the entanglement measurement through the detection in complementary local bases by rotating the polarizers P2 and P3, and the wave plates HWP5 and quarter-wave plate 2. Due to the different incident directions of the pump pulses at the upper and the lower paths, the corresponding phonon modes excited in the diamond have different momenta, so they represent independent modes even if they have partial spatial overlap. The phonon is converted to the anti-Stokes photon by the retrieval pulse, so we measure the photon–phonon state by detecting the coincidence counts between Stokes and anti-Stokes photons in different bases. In Fig. 3a, we show the registered coincidence counts, as we rotate the angle of the polarizer P2. The oscillation of the coincidence counts with a visibility of $(74.6 \pm 3.6)\%$ is an indicator of coherence of the underlying state. To verify

entanglement of the photon–phonon state, we use quantum state tomography to reconstruct the full density matrix from the measured coincidence counts[31], with the resulting matrix elements shown in Fig. 3b. From the reconstructed density matrix $\rho_e$, we find its entanglement fidelity, defined as the maximum overlap of $\rho_e$ with a maximally entangled state, $F_e = (81.0 \pm 1.8)\%$, significantly higher than the criterion of $F_e = 0.5$ for verification of entanglement[32]. The error bars are determined by assuming a Poissonian distribution for the photon counts and propagated from the raw data to the calculated quantities through exact numerical simulation. The dominant noise in this system comes from the accidental coincidence between the detected Stokes and the anti-Stokes photons[20,21]. To measure the contribution of this accidental coincidence, we introduce an extra time delay of 13 ns, the repetition period of our pump pulses, to one of the detectors when we record the coincidence. When we subtract the background noise due to this accidental coincidence, the resulting matrix elements reconstructed from the quantum state tomography are shown in Fig. 3c. We find the entanglement fidelity is improved to $F_e = (89.7 \pm 1.2)\%$ after subtraction of the accidental coincidence.

To perform quantum teleportation using the photon–phonon entanglement, we first transform the effective photon–phonon-entangled state to the standard form of equation 2, by the semicircle HWPs in the state preparation box of Fig. 2. The polarizer P1, and the wave plates HWP2 and quarter-wave plate 1 then prepare the to-be-teleported photon polarization to arbitrary superposition states $|\Phi_{in}\rangle = c_0|H\rangle_t + c_1|V\rangle_t$. We perform Bell

measurement through the calcite C2, the HWP3, the polarizer P2 and the detector APD2. For instance, with the HWP3 set at 0° and the polarizer P2 set along the direction $|H\rangle + |V\rangle$, a photon count in the detector APD2 corresponds to a projection to the Bell state $(|H\rangle_t|U\rangle_t + |V\rangle_t|L\rangle_t)/\sqrt{2}$ for the polarization and the path qubits of the photon before the measurement box. By rotating the angles of HWP3 and P2, we can also perform projection to any other Bell states.

The experimental result for teleportation is shown in Fig. 4. The teleportation fidelity is defined as $F = \langle\Phi_{in}|\rho_{out}|\Phi_{in}\rangle$, where $|\Phi_{in}\rangle$ is the input state at Alice's side and $\rho_{out}$ denotes the output density matrix at Bob's side, reconstructed through quantum state tomography measurements. In Fig. 4a, we show the teleportation fidelity under six complementary bases states with $|\Phi_{in}\rangle = |H\rangle_t$, $|V\rangle_t, |\pm\rangle_t = (|H\rangle_t \pm |V\rangle_t)/\sqrt{2}$, $|L\rangle_t = (|H\rangle_t + i|V\rangle_t)/\sqrt{2}, |R\rangle_t = (|H\rangle_t - i|V\rangle_t)/\sqrt{2}$ in cases with and without subtraction of the background noise. The average fidelity over these six bases states is $(93.9 \pm 0.8)\%$ (or $(83.0 \pm 0.8)\%$) with (or without) background noise subtraction. This average fidelity is significantly $>2/3$, the boundary value for the fidelity that separates quantum teleportation from classical operations. For more complete characterization, we also perform QPT for the teleportation operation. In the ideal case, teleportation should be characterized by an identity transformation, meaning that Alice's input state is teleported perfectly to Bob's side. The experimentally reconstructed process matrix elements are shown in Fig. 4b (see Methods for explanation of QPT). The process fidelity is given by $F_p = (85.9 \pm 1.6)\%$, which corresponds to a teleportation fidelity $\bar{F} = (90.6 \pm 1.0)\%$ averaged over all possible input states, with equal weight in the qubit space.

## Discussion

Teleportation of the quantum states from a photon to the vibration modes of a millimeter-sized diamond under ambient conditions generates a quantum link between the microscopic particle and the macroscopic world around us, usually under the law of classical physics. In our experiment, the ultrafast laser technology provides the key tool for the fast processing and detection of quantum states within its short life time in macroscopic objects, consisting of many strongly interacting atoms that are coupled to the environment. Combined with the tunability of the wavelength for the retrieval laser pulse[23], the technique introduced in our experiment would be useful for the realization of a new source of entangled photons based on the diamond optomechanical coupling with the dual-rail encoding. Such a source could generate entangled photons at wavelengths inconvenient to produce by other methods. For instance, we may generate entanglement between the ultraviolet and infrared photons, with the infrared photon good for quantum communication and the ultraviolet photon convenient to be interfaced with other qubits, such as the ion matter qubits. Such a photon source is hard to generate by the conventional spontaneous parametric down conversion method. In future, the tools based on the ultrafast pump and probe could be combined with the powerful laser cooling or low-temperature technology to provide more efficient ways for quantum control of the optomechanical systems, with important applications for realization of transduction of quantum signals[25,26], processing of quantum information or single-photon signals[19,20,23] and sensing of small mechanical vibrations[19].

## Methods

**Quantum process tomography.** QPT[31] is defined by a completely positive map $\varepsilon$: $\rho_f \equiv \varepsilon(\rho_i)$ that transfers an arbitrary input state $\rho_i$ to the output $\rho_f$. It can be characterized by a unique process matrix $\chi_{mn}$ through the map

$\rho_f = \sum_{mn} E_m \rho_i E_n^\dagger \chi_{mn}$, by choosing a fixed set of basis operator $E_m$. In our experiment, we set the basis operators $E_m$ to be the identity operator $I$ and the three Pauli matrices $X = \sigma_x, Y = -i\sigma_y, Z = \sigma_z$. This corresponds to a choice of six complementary input states $|H\rangle, |V\rangle, |+\rangle, |-\rangle, |L\rangle$ and $|R\rangle$ for the teleportation. We reconstruct the output state from teleportation by quantum state tomography and use them to calculate the process matrix $\chi$ through the maximally likelihood estimation[31]. The process fidelity is determined by $F_p = Tr(\chi\chi_{id})$, where $\chi_{id}$ is the identity process matrix corresponding to the perfect case. The process fidelity $F_p$ determines the average teleportation fidelity $\bar{F}$ by the formula $\bar{F} = (2F_p + 1)/3$ (ref. 31), where $\bar{F}$ is defined as the fidelity averaged over all possible states of the input qubit with equal weight.

**Data availability.** The data that support the findings of this study are available from the corresponding author upon request.

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

## Acknowledgements

This work was supported by the Ministry of Education of China through its grant to Tsinghua University. L.M.D. acknowledges in addition support from the Intelligence Advanced Research Projects Activity (IARPA) quantum computing program, the Army Research Lab (ARL) quantum network program, and the Air Force Office of Scientific Research (AFOSR) Multidisciplinary University Research Initiative (MURI) program.

## Author contributions

L.M.D. designed the experiment and supervised the project. P.Y.H., Y.Y.H., X.X.Y., X.Y.C., C.Z. and L.H. carried out the experiment. L.M.D. and P.Y.H. wrote the manuscript.

## Additional information

**Competing financial interests:** The authors declare no competing financial interests.

