## [Peer Review File · Nature Communications]

Reviewer #1 (Remarks to the Author):

"Quantum teleportation from light beams to vibrational states of a macroscopic diamond" is an interesting paper that utilizes the phonon states in bulk diamond to perform quantum processing at room temperature. The main claim of the paper is that there is demonstrated quantum state teleportation from light to phonons.

The paper is novel, and broadly interesting, and I would likely recommend for publication after a few clarifications.

-The claim that there is teleportation between the light states and phonon states is a bit too bold. At the time of the measurement - maybe a few nanoseconds after read/write - both the Stokes and anti-Stokes light has departed the diamond and the phonon has long decayed. How can there be teleportation to the phonon if it is already decayed? It seems that the primary use of the diamond is to construct two-colour mode entanglement, which is then indeed teleported. I believe then that this point needs to be clarified and may require a new title

-There is overlap of the two modes within the diamond. To what extent is this possible without mixing the two modes? What is the geometry of the focus like? Ie, what is the size of the non-classical state in the phonon?

-Describe the diamond in more detail. Crystal axes? Mono crystalline? Source? etc?

-The manuscript builds on previous work investigating the recent developments in diamond quantum optics. It would be worthwhile to spend a little more time putting the the submitted manuscript in context of existing developments in the area. I would suggest citing the following additional works and clarifying the increment that has been achieved with your submission:

<http://arxiv.org/abs/1509.05098>

<http://arxiv.org/abs/1508.01729>

Reply to referee #1

Comment: "I am satisfied with the response and recommend publication.

I would advise that before publication two references in the manuscript might be updated now that they are published:

-"[23] Fisher, K. A. G. et al. Quantum optical signal processing in diamond. Preprint at <http://arxiv.org/abs/1509.05098>." is now published at Nature Communications 7, 11200, 2016. (<http://www.nature.com/ncomms/2016/160405/ncomms11200/full/ncomms11200.html>)

-"[24] Bustard, P. J. et al. Ultrafast slow-light: Raman- induced delay of THz-bandwidth pulses. Preprint at <http://arxiv.org/abs/1508.01729>." is now published at Phys. Rev. A 93, 043810 (<https://journals.aps.org/pr/abstract/10.1103/PhysRevA.93.043810>)."

Reply: We thank the referee for recommendation of publication of this manuscript. We have updated the references [23] and [24] to their published versions.

-Typo: "Ti-Sapphire aser (Coherent Verdi 18)": Verdi is not Ti:Sapphire.

-It would be valuable to share some click rates for tomography plots.

Overall the paper is an exciting development in the area of diamond phonon quantum optics. It is an increasing demonstration that, despite the challenges of a short decay time, there are many new opportunities for the diamond phonon platform. However, before publication, I would suggest the clarifications above.

Reviewer #2 (Remarks to the Author):

The results reported by Prof. Duan and co-workers describe a quantum teleportation experiment between photons and macroscopic phonon modes inside a diamond. It builds up on previous results of coherent control and entanglement of the phonon modes in diamond by Prof. Walmsley's group (ref. 20 and 21). In my opinion the results are an important step towards pushing quantum effects to more and more macroscopic structures. The manuscript also reports an interesting approach to frequency conversion based on their teleportation scheme. I believe that the results are suited for publication in nature communications if the following points are satisfactorily addressed.

1) On page 1 abstract 2 the authors state that the entanglement reported in ref. 20 is not suited for teleportation and that the entanglement reported here is different. However, the mechanism for creating entanglement and also the setup appears to be very much the same. In ref 20 two separated diamonds are being used each hosting a phonon mode for the entanglement whereas in the present work one diamond is hosting two separate phononic modes. If a stokes photon is detected, after the two optical paths behind the diamond(s) are overlapped on a beamsplitter, the resulting state between the phonon modes would be the same for ref 20 and the reported work. I think the difference is that in ref 20 the entanglement between the two phonon modes is being discussed whereas in the present work entanglement between the phonon mode and the stokes photon is being used for the teleportation. I therefore think the authors need to clarify their comparison.

2) To put this work into context of other experimental realizations of teleportation I would find it helpful to clarify what specific kind of teleportation is being implemented. This is in particular important for readers that are less familiar to the subject. In the presented case a deterministic bell state measurement cannot be performed and the teleportation events are post-selected on a successful measurement outcome. Furthermore the entangled state described in equation (2) only exists conditional on the presence of a stokes photon which can only be verified by detecting the photon and thereby destroying the entangled state. And third, the phonon state is already read out before the Bell state measurement is performed meaning before the teleportation is completed. Following van Enk et al. Physical Review A 75, 052318 (2007) the teleportation reported appears to be a-posteriori conditional teleportation.

3) On page 3 paragraph 2 the authors discuss the fact that the phonon state is readout before the teleportation is completed. They compare this to delayed-choice quantum experiments. I find this comparison misleading and suggest to leave it out. In delayed-choice experiments one tries to test the particle wave duality whereas in a teleportation experiment one tries to faithfully transmit a quantum state - the 'spirit' of the experiments is very different.

4) I think the reader would benefit from seeing the 'raw-data' of the experiment. The bar plots in figure 3 and 4 are probably inferred from coincidence measurements between the 2 photon detectors while varying the position of the wave plates. An example curve of this would give the reader a better idea on how the measurement is carried out.

5) In the caption for figure one (last sentence) the authors state that 'through classical communication, the phonon state is projected to the same input on Alice side'. I find this unclear. In deterministic quantum teleportation the input state is transferred by a quantum measurement (the bell state measurement) and the classical communication of it's outcome to Bob's side and Bob performing a feedback according to this outcome. This is not the case here (see above in 2)).

Reviewer #3 (Remarks to the Author):

Hou et al. demonstrate a quantum teleportation protocol for the transfer of quantum states of single photons to vibrational states of diamond crystal at room temperature. This result adds to the growing body of work demonstrating high-level quantum control of mechanical degrees of freedom

in solid state systems. Optical phonon modes in diamond crystals have been used in previous experiments by other researchers to generate nonclassical quantum states of phonons and to entangle phonon modes in two spatially separate diamonds. This latter experiment is based on an entanglement swapping procedure, that is, on a light-to-matter teleportation protocol where the photon state to be teleported itself is entangled with a phonon mode. Thus, the present paper is not the first to demonstrate photon-to-phonon teleportation (which is also not claimed by the authors). The main improvement demonstrated in the present work is the use of a dual-rail encoding (as opposed to the single-rail encoding used in previous experiment) which uses two phonon (or photon) modes to encode the state of one qubit. This is an important step as it provides a richer and more efficient toolbox for manipulation and measurement of quantum states. This development parallels the one which occurred in experiments with atomic ensembles interacting with light in the context of quantum repeater protocols. Hou et al. thus report the first photon-to-phonon teleportation using dual-rail encoding. The experiment relies on two techniques which heavily simplify the experimental procedure: Firstly, the Bell measurement on two incoming photons - required in a fully-fledged quantum teleportation - is avoided by encoding two qubits in one photon and performing a suitable detection on this single photon. Similar techniques have been used in experiments with atomic ensembles for proof-of-principle demonstrations bypassing the difficulty of an interferometric measurement. Secondly, and more importantly, in a "reversed time ordering" the phonon state is read out and mapped to a second pulse of light before the first photon is actually measured. This is necessary because of the very short quantum coherence time of the phononic states. Therefore, the experiment could be equally well be interpreted as demonstrating (i) a new source of entangled photons and (ii) a photon-to-photon teleportation based on such an entangled state.

Overall, I think the article presents good work and demonstrates an important technical advance over previous work. The data presented and the explanations are convincing. The presentation is good modulo some issues with the English (E.g. "A prominent example is provided by mechanical vibration in macroscopic solids [...] where quantum coherence is hard to survive."). My main concern pertains to the perspective and implication of this work: Light-to-matter teleportation is an important sub-routine in quantum communication and quantum repeater protocols. However, as the authors themselves emphasize, "the reversed time ordering in this demonstration of quantum teleportation makes it unsuitable for application in quantum repeaters which requires a much longer memory time". Beyond applications, one can argue (as the authors do) that experiments with mechanical oscillators offer tests of quantum physics at macroscopic scales. But in this regard I do not see where the present experiment goes beyond what has been demonstrated in previous ones with optical phonons in diamond. Therefore I am not convinced the demonstrated quantum teleportation "is of both fundamental interest and practical importance". The point of this experiment which is most intriguing to me is the tunability of the wavelength of the entangled photon pairs. The authors mention this aspect of their new source of entangled photons but it remains unclear to me how important and efficient this process can be for applications in quantum technologies. While I think the paper is correct and certainly provides publishable results, I am not sure its perspective and implications warrant publication in Nature Communications.

Reply to referees

We thank all the referees for their recommendation, insightful comments, and helpful suggestions to improve the manuscript. We have followed the referees' suggestions to revise the manuscript and carefully addressed all the comments raised by the referees. In the following, let us address the referees' comments point by point:

Reply to referee #1

Comment: "Quantum teleportation from light beams to vibrational states of a macroscopic diamond" is an interesting paper that utilizes the phonon states in bulk diamond to perform quantum processing at room temperature. The main claim of the paper is that there is demonstrated quantum state teleportation from light to phonons.

The paper is novel, and broadly interesting, and I would likely recommend for publication after a few clarifications."

Reply: Thank the referee for the recommendation.

Comment: "-The claim that there is teleportation between the light states and phonon states is a bit too bold. At the time of the measurement - maybe a few nanoseconds after read/write - both the Stokes and anti-Stokes light has departed the diamond and the phonon has long decayed. How can there be teleportation to the phonon if it is already decayed? It seems that the primary use of the diamond is to construct two-colour mode entanglement, which is then indeed teleported. I believe then that this point needs to be clarified and may require a new title"

Reply: We thank the referee for bringing up this important point that needs to be further clarified. In this paper, we have followed the language and the method of Ref. [20] (the pioneering diamond phonon quantum optics paper published in Science 334, 1253-1256 (2011)) in using "the reversed time ordering" trick to measure the phonon modes in advance to overcome the issue associated with their short life times. Same as Ref. [20], which measures the phonon's state of the diamond before the entangling operation, here we read out the photon's state before the teleportation protocol is completed. As mentioned in Ref. [20], the conventional time ordering could be recovered on diamond quantum chips using chip-integrated fast detectors. For bulk diamond material, measuring in advance with the reversed time ordering is a useful trick to save the phonon's state from its fast decay with the understanding that the phonon-to-photon mapping does not change the character of the underlying state.

To clarify this point, in the revised version, we have mentioned explicitly in the introduction of the paper that our implementation of teleportation involves the reversed time ordering where the phonon states are read out before the teleportation is completed. In the text, we have further discussed the consequence and limitation that the implementation with the reversed time ordering could bring up to its potential applications. Note that for some application, such as for quantum frequency conversion, the reversed time ordering in the implementation is not a problem at all, and teleportation gives a convenient picture to understand this application, where the teleported

phonon's state is read out by a light beam at any desired frequency, converting the photon's frequency from one to another without changing its quantum state. With this clarification and discussion, and with its conformity to the language for research in this direction (such as ref. [20]), we believe the terminology used here would not cause confusion or misunderstanding any more.

Comment: "--There is overlap of the two modes within the diamond. To what extent is this possible without mixing the two modes? What is the geometry of the focus like? Ie, what is the size of the non-classical state in the phonon?"

Reply: The two phonon modes in the diamond (representing the qubit basis states) are excited by the pump laser beams of orthogonal polarizations and therefore they are independent modes in the diamond with orthogonal polarizations. Even when these two modes have some overlap in space, there is no mode mixing because of orthogonality and selection rules in polarization. We have clarified this point in the revised version.

Comment: "--Describe the diamond in more detail. Crystal axes? Mono crystalline? Source? etc?"

Reply: The diamond is a type IIa, single-crystal synthetic diamond sample from the Element Six company, with size of 3mm*3mm*0.3mm, cut along the 100 face. We have clarified this in the revised version.

Comment: "--The manuscript builds on previous work investigating the recent developments in diamond quantum optics. It would be worthwhile to spend a little more time putting the submitted manuscript in context of existing developments in the area. I would suggest citing the following additional works and clarifying the increment that has been achieved with your submission:

<http://arxiv.org/abs/1509.05098>

<http://arxiv.org/abs/1508.01729>"

Reply: Thank the referee for bringing to our attention the recent related works in this direction. We have cited and related to these works in the revised version.

Comment: "--Typo: "Ti-Sapphire aser (Coherent Verdi 18)": Verdi is not Ti:Sapphire.

-It would be valuable to share some click rates for tomography plots."

Reply: We have corrected the typos and added the click rates for the tomography plots. In the tomography measurements, the coincidence count rate for Stokes and anti-Stokes photons is 8 per second for measurements in the UU and LL bases. We have also added a new Fig. 3a, which show the raw data (the oscillation of the coincidence counts as a function of the polarization rotation angle) as an indicator of coherence of the underlying quantum state.

Comment: "Overall the paper is an exciting development in the area of diamond phonon quantum optics. It is an increasing demonstration that, despite the challenges of a short decay time, there are many new opportunities for the diamond phonon platform. However, before publication, I would suggest the clarifications above."

Reply: We thank the referee again for the enthusiastic recommendation and have made all the clarifications pointed out by the referee.

Reply to referee #2

Comment: “The results reported by Prof. Duan and co-workers describe a quantum teleportation experiment between photons and macroscopic phonon modes inside a diamond. It builds up on previous results of coherent control and entanglement of the phonon modes in diamond by Prof. Walmsley's group (ref. 20 and 21). In my opinion the results are an important step towards pushing quantum effects to more and more macroscopic structures. The manuscript also reports an interesting approach to frequency conversion based on their teleportation scheme. I believe that the results are suited for publication in nature communications if the following points are satisfactorily addressed. “

Reply: Thank the referee for the recommendation.

Comment: “1) On page 1 abstract 2 the authors state that the entanglement reported in ref. 20 is not suited for teleportation and that the entanglement reported here is different. However, the mechanism for creating entanglement and also the setup appears to be very much the same. In ref 20 two separated diamonds are being used each hosting a phonon mode for the entanglement whereas in the present work one diamond is hosting two separate phononic modes. If a stokes photon is detected, after the two optical paths behind the diamond(s) are overlapped on a beamsplitter, the resulting state between the phonon modes would be the same for ref 20 and the reported work. I think the difference is that in ref 20 the entanglement between the two phonon modes is being discussed whereas in the present work entanglement between the phonon mode and the stokes photon is being used for the teleportation. I therefore think the authors need to clarify their comparison.”

Reply: We have revised and clarified the comparison with Ref. [20] to make it more accurate. As pointed out by the referee, Ref. [20] is for entanglement between two phonon modes in separated diamonds using the single-rail encoding with one phonon mode per diamond. Here we implementation teleportation using phonon-photon entanglement in the dual-rail representation (two phonon modes per diamond to represent a qubit).

Comment: “2) To put this work into context of other experimental realizations of teleportation I would find it helpful to clarify what specific kind of teleportation is being implemented. This is in particular important for readers that are less familiar to the subject. In the presented case a deterministic bell state measurement cannot be performed and the teleportation events are post-selected on a successful measurement outcome. Furthermore the entangled state described in equation (2) only exists conditional on the presence of a stokes photon which can only be verified by detecting the photon and thereby destroying the entangled state. And third, the phonon state is already read out before the Bell state measurement is performed meaning before the teleportation is completed. Following van Enk et al. Physical Review A 75, 052318 (2007) the teleportation reported appears to be a-posteriori conditional teleportation.”

Reply: Following the referee's suggestion, we have clarified in the introduction of the revised manuscript that the implementation of teleportation here is conditional based on post-selecting of successful measurement outcomes, similar to the pioneering experiment on photon teleportation in Refs. [5,6], and this implementation has a reversed time ordering where the phonon's state is measured in advance similar to the pioneering diamond phonon entanglement experiment in Ref. [20].

Comment: “3) On page 3 paragraph 2 the authors discuss the fact that the phonon state is readout before the teleportation is completed. They compare this to delayed-choice quantum experiments. I find this comparison misleading and suggest to leave it out. In delayed-choice experiments one tries to test the particle wave duality whereas in a teleportation experiment one tries to faithfully transmit a quantum state - the 'spirit' of the experiments is very different.”

Reply: Following the referee's suggestion, we have left out the comparison with the delayed choice experiments in the revised version

Comment: “4) I think the reader would benefit from seeing the 'raw-data' of the experiment. The bar plots in figure 3 and 4 are probably inferred from coincidence measurements between the 2 photon detectors while varying the position of the wave plates. An example curve of this would give the reader a better idea on how the measurement is carried out.”

Reply: Following the referee's suggestion, we have added a new Fig. 3a, which shows the raw data (the oscillation of the coincidence counts as a function of the rotation angle of the polarizer) as an indicator of coherence of the quantum state.

Comment: “5) In the caption for figure one (last sentence) the authors state that 'through classical communication, the phonon state is projected to the same input on Alice side'. I find this unclear. In deterministic quantum teleportation the input state is transferred by a quantum measurement (the bell state measurement) and the classical communication of it's outcome to Bob's side and Bob performing a feedback according to this outcome. This is not the case here (see above in 2)).”

Reply: Thanks for pointing this out. We have corrected this sentence to “Conditional on certain measurement outcomes, the phonon state is projected to the same input on Alice side”

Reply to referee #3

Comment: “Hou et al. demonstrate a quantum teleportation protocol for the transfer of quantum states of single photons to vibrational states of diamond crystal at room temperature. This result adds to the growing body of work demonstrating high-level quantum control of mechanical degrees of freedom in solid state systems. Optical phonon modes in diamond crystals have been used in previous experiments by other researchers to generate nonclassical quantum states of phonons and to entangle phonon modes in two spatially separate diamonds. This latter experiment is based on an entanglement swapping procedure, that is, on a light-to-matter teleportation protocol where the photon state to be teleported itself is entangled with a phonon mode. Thus, the present paper is not the first to demonstrate photon-to-phonon teleportation (which is also not

claimed by the authors). The main improvement demonstrated in the present work is the use of a dual-rail encoding (as opposed to the single-rail encoding used in previous experiment) which uses two phonon (or photon) modes to encode the state of one qubit. This is an important step as it provides a richer and more efficient toolbox for manipulation and measurement of quantum states. This development parallels the one which occurred in experiments with atomic ensembles interacting with light in the context of quantum repeater protocols. Hou et al. thus report the first photon-to-phonon teleportation using dual-rail encoding. The experiment relies on two techniques which heavily simplify the experimental procedure: Firstly, the Bell measurement on two incoming photons -required in a fully-fledged quantum teleportation - is avoided by encoding two qubits in one photon and performing a suitable detection on this single photon. Similar techniques have been used in experiments with atomic ensembles for proof-of-principle demonstrations bypassing the difficulty of an interferometric measurement. Secondly, and more importantly, in a "reversed time ordering" the phonon state is read out and mapped to a second pulse of light before the first photon is actually measured. This is necessary because of the very short quantum coherence time of the phononic states. Therefore, the experiment could be equally well be interpreted as demonstrating (i) a new source of entangled photons and (ii) a photon-to-phonon teleportation based on such an entangled state."

Reply: We thank the referee for the recommendation and helpful comments. The referee brings up an interesting question about the interpretation of Ref. [20] also as a photon-to-phonon teleportation experiment. This point certainly needs to be further clarified. In Ref. [20], after the laser pumping of the diamond the state of the photon and the phonon is entangled in the number basis with the following form

$$|\varphi_{nt}\rangle = \sqrt{1-p}|00\rangle + \sqrt{p}|11\rangle \quad (1)$$

In this equation $|11\rangle$ means that we have one photon and one phonon excitation and we have neglected small higher order excitation terms. The excitation probability p is required to be much less than 1 (say, around 1%) for this kind of protocol to work. Now with an input photon state $|\varphi\rangle = c_0|0\rangle + c_1|1\rangle$ with unknown coefficients, the measurement there, conditional on a successful registration of one photon click, effectively projects the photon state before the detection to the EPR state $|EPR\rangle = |01\rangle + |10\rangle$ (unnormalized and neglecting a possible phase factor). The phonon state after this projection is given by $\langle EPR|\varphi\rangle|\varphi_{nt}\rangle = c_1\sqrt{1-p}|0\rangle + c_0\sqrt{p}|1\rangle$. Note that as $p \ll 1$ in Eq. (1), the output phonon state is very different from the input state $|\varphi\rangle$ for the photon, and thus the operation in the experiment [20] is different from the conventional teleportation protocol as proposed in Ref. [1], even conditional on the successful measurement event. Similarly, if the input photon is in an entangled state with another phonon $|\varphi\rangle = c_0|00\rangle + c_1|11\rangle$, after the projection measurement, the phonon state of the two diamonds has the form $c_1\sqrt{1-p}|01\rangle + c_0\sqrt{p}|10\rangle$, which is different from the input entangled state (and thus the operation is different from the conventional entanglement swapping protocol which preserves the input entangled state). The purpose of the experiment in Ref. [20], as emphasized there, is to generate an entangled state of the two diamonds by fixing $c_1 = \sqrt{p}$, instead of demonstrating teleportation of unknown states or entanglement swapping. The dual-rail encoding in our experiment, as pointed out by the referee, gives a convenient demonstration of teleportation.

Comment: “Overall, I think the article presents good work and demonstrates an important technical advance over previous work. The data presented and the explanations are convincing. The presentation is good modulo some issues with the English (E.g. "A prominent example is provided by mechanical vibration in macroscopic solids [...] where quantum coherence is hard to survive."). My main concern pertains to the perspective and implication of this work: Light-to-matter teleportation is an important sub-routine in quantum communication and quantum repeater protocols. However, as the authors themselves emphasize, "the reversed time ordering in this demonstration of quantum teleportation makes it unsuitable for application in quantum repeaters which requires a much longer memory time". Beyond applications, one can argue (as the authors do) that experiments with mechanical oscillators offer tests of quantum physics at macroscopic scales. But in this regard I do not see where the present experiment goes beyond what has been demonstrated in previous ones with optical phonons in diamond. Therefore I am not convinced the demonstrated quantum teleportation "is of both fundamental interest and practical importance". The point of this experiment which is most intriguing to me is the tunability of the wavelength of the entangled photon pairs. The authors mention this aspect of their new source of entangled photons but it remains unclear to me how important and efficient this process can be for applications in quantum technologies. While I think the paper is correct and certainly provides publishable results, I am not sure its perspective and implications warrant publication in Nature Communications.”

Reply: As discussed in the reply above, on the conceptual side, this experiment is quite different from Ref. [20], although both of the experiments used the “reversed time ordering” trick in the demonstration. Ref. [20] focused on demonstration of entanglement generation between two macroscopic diamonds, while our experiment demonstrated teleportation involving vibration of a macroscopic diamond. Given the broad interest in teleportation towards more macroscopic objects, we believe this experiment represents “an important step towards pushing quantum effects to more and more macroscopic structures”, as emphasized by the other referees. On the application side, although the teleportation here wouldn’t be used for quantum repeaters due to the short life time of the diamond phonons, it indeed opens up the perspective to realize new source of entangled photons with tunability in wavelength and to implement a quantum frequency transducer (converter). Note that the entangled photon source plays a very important role in quantum information processing. The technique introduced in this experiment would be useful for realization of a new source of entangled photons based on the diamond optomechanical coupling with the dual-rail encoding. Such a source could generate entangled photons at wavelengths inconvenient to produce by other methods. For instance, we may generate entanglement between ultraviolet and infrared photons, with the infrared photon good for quantum communication and the ultraviolet photon convenient to be interfaced with other qubits, such as the ion matter qubits. Such a photon source is hard to be generated by the conventional spontaneous parametric down conversion method. Quantum frequency transducer has been recognized as a critical component of a quantum network (see, e.g., Ref. [25,26]). For instance, frequency conversion for a weak classical light using optomechanical coupling has been published recently in Science 338, 1609 (2012). Our teleportation experiment opens up the perspective to realize a quantum frequency transducer for a single photon to any desired frequency with the optomechanical coupling while preserving its polarization quantum state, which has important application potentials.

In the revised version, we have further clarified the advance represented by this experiment and discussed more clearly the application perspective opened up by this experiment. We have also improved English and presentation as pointed out by the referee. With this improvement, we believe this work, as an important advance in this area, is of broad interest and suitable for publication in Nature Communications.

Reviewers' comments:

Reviewer #1 (Remarks to the Author):

In their response the author's write:

"The two phonon modes in the diamond (representing the qubit basis states) are excited by the pump laser beams of orthogonal polarizations and therefore they are independent modes in the diamond with orthogonal polarizations. Even when these two modes have some overlap in space, there is no mode mixing because of orthogonality and selection rules in polarization. We have clarified this point in the revised version."

As I understand for the experimental geometry, the two polarizations couple to the same LO phonon mode. That is, for incidence on $\langle 100 \rangle$ both H and V polarized light will couple to the same matter state. The diamond crystal group has a triply degenerate mode, but I understand that for the geometry used both H and V couple to the same LO mode. That would imply that what is written in the response is incorrect. I would request that the authors confirm the group theory. If their statement is indeed correct I would suggest acceptance of the manuscript.

Reviewer #2 (Remarks to the Author):

I am satisfied with the answers and revisions done by the authors and would now recommend the manuscript for publication.

Reviewer #3 (Remarks to the Author):

The authors carefully addressed the concerns raised in the three reports. In particular, they satisfactorily explain the difference between their work and the one presented in Ref. [20]. In the revised manuscript the experiment is better put into context regarding the aspect of frequency conversion. Overall, I am now much more convinced of the novelty and significance of the presented work. I recommend acceptance of the manuscript in its present form.

One last remark: The authors might want to reconsider their wording concerning the "trick of reversed time ordering introduced in Ref. [20]". The authors of [20] do not use this wording. I believe it would be more appropriate to speak of a "technique" instead of a "trick".

Reply to referee #1

Comment: "I have read the revised response and only have one additional comment. It is small, but there may be an error so I would advise making sure to correct before acceptance.

In their response the author's write:

"The two phonon modes in the diamond (representing the qubit basis states) are excited by the pump laser beams of orthogonal polarizations and therefore they are independent modes in the diamond with orthogonal polarizations. Even when these two modes have some overlap in space, there is no mode mixing because of orthogonality and selection rules in polarization. We have clarified this point in the revised version."

As I understand for the experimental geometry, the two polarizations couple to the same LO phonon mode. That is, for incidence on $\langle 100 \rangle$ both H and V polarized light will couple to the same matter state. The diamond crystal group has a triply degenerate mode, but I understand that for the geometry used both H and V couple to the same LO mode. That would imply that what is written in the response is incorrect. I would request that the authors confirm the group theory. If their statement is indeed correct I would suggest acceptance of the manuscript."

Reply: We thank the referee for recommendation of acceptance of the manuscript after a small correction. The referee brought up an interesting point which should be clarified and corrected. The referee is right that the pump beams of different polarizations could couple to the phonon mode of the same polarization. However, in our experiment the two phonon modes excited by different pump beams are still independent because of their different momenta. The two pump beams are incident to the diamond from different directions after the lens, and due to the momentum conservation (or the phase matching condition), the excited phonon carries the momentum difference between the pump beam and the Stokes photon. This momentum difference is significantly different for the two incident directions, which distinguishes the corresponding phonon modes. We have experimentally seen that the phonon excited by the write beam in one direction cannot be retrieved by the readout beam in the other direction, which demonstrated the independence of the corresponding phonon modes.

We have clarified the statement in the revised manuscript as "Due to the different incident directions of the pump pulses at the upper and the lower paths, the corresponding phonon modes excited in the diamond have different momenta, so they represent independent modes even if they have partial spatial overlap".

Reply to referee #2

Comment: "I am satisfied with the answers and revisions done by the authors and would now recommend the manuscript for publication."

Reply: Thank the referee for recommendation of the manuscript for publication.

Reply to referee #3

Comment: “The authors carefully addressed the concerns raised in the three reports. In particular, they satisfactorily explain the difference between their work and the one presented in Ref. [20]. In the revised manuscript the experiment is better put into context regarding the aspect of frequency conversion. Overall, I am now much more convinced of the novelty and significance of the presented work. I recommend acceptance of the manuscript in its present form.

One last remark: The authors might want to reconsider their wording concerning the "trick of reversed time ordering introduced in Ref. [20]". The authors of [20] do not use this wording. I believe it would be more appropriate to speak of a "technique" instead of a "trick".”

Reply: We thank the referee for recommendation of the manuscript for publication. We have improved the wording of that sentence and changed “trick” to the more appropriate word “technique”.

Reviewer #1 (Remarks to the Author):

I am satisfied with the response and recommend publication.

I would advise that before publication two references in the manuscript might be updated now that they are published:

-"[23] Fisher, K. A. G. et al. Quantum optical signal processing in diamond. Preprint at <http://arxiv.org/abs/1509.05098>."

is now published at Nature Communications 7, 11200, 2016.
(<http://www.nature.com/ncomms/2016/160405/ncomms11200/full/ncomms11200.html>)

-"[24] Bustard, P. J. et al. Ultrafast slow-light: Raman-induced delay of THz-bandwidth pulses. Preprint at <http://arxiv.org/abs/1508.01729>."

is now published at Phys. Rev. A 93, 043810
(<https://journals.aps.org/pr/abstract/10.1103/PhysRevA.93.043810>)